# MULTI-AGENT MDP HOMOMORPHIC NETWORKS

**Elise van der Pol**
UvA-Bosch Deltalab
University of Amsterdam
e.e.vanderpol@uva.nl

**Herke van Hoof**
UvA-Bosch Deltalab
University of Amsterdam
h.c.vanhoof@uva.nl

**Frans A. Oliehoek**
Department of Intelligent Systems
Delft University of Technology
f.a.oliehoek@tudelft.nl

**Max Welling**
UvA-Bosch Deltalab
University of Amsterdam
m.welling@uva.nl

## ABSTRACT

This paper introduces Multi-Agent MDP Homomorphic Networks, a class of networks that allows distributed execution using only local information, yet is able to share experience between global symmetries in the joint state-action space of cooperative multi-agent systems. In cooperative multi-agent systems, complex symmetries arise between different configurations of the agents and their local observations. For example, consider a group of agents navigating: rotating the state globally results in a permutation of the optimal joint policy. Existing work on symmetries in single agent reinforcement learning can only be generalized to the fully centralized setting, because such approaches rely on the global symmetry in the full state-action spaces, and these can result in correspondences across agents. To encode such symmetries while still allowing distributed execution we propose a factorization that decomposes global symmetries into local transformations. Our proposed factorization allows for distributing the computation that enforces global symmetries over local agents and local interactions. We introduce a multi-agent equivariant policy network based on this factorization. We show empirically on symmetric multi-agent problems that globally symmetric distributable policies improve data efficiency compared to non-equivariant baselines.

## 1 INTRODUCTION

Equivariant and geometric deep learning have gained traction in recent years, showing promising results in supervised learning (Cohen & Welling, 2016; Winkels & Cohen, 2018; Weiler et al., 2018; Weiler & Cesa, 2019; Worrall et al., 2017; Fuchs et al., 2020; Thomas et al., 2018), unsupervised learning (Dey et al., 2021) and reinforcement learning (van der Pol et al., 2020; Simm et al., 2021). In single agent reinforcement learning, enforcing equivariance to group symmetries has been shown to improve data efficiency, for example with MDP homomorphic networks (van der Pol et al., 2020), trajectory augmentation (Lin et al., 2020; Mavalankar, 2020), or symmetric locomotion policies (Abdolhosseini et al., 2019). Equivariance enables an agent to learn policies more efficiently within its environment by sharing weights between state-action pairs that are equivalent under a transformation. As a result of this weight sharing, the agent implicitly learns a policy in a reduced version of the MDP. We are interested in cooperative multi-agent reinforcement learning, where symmetries exist both in the global environment, and between individual agents in the larger multi-agent system.

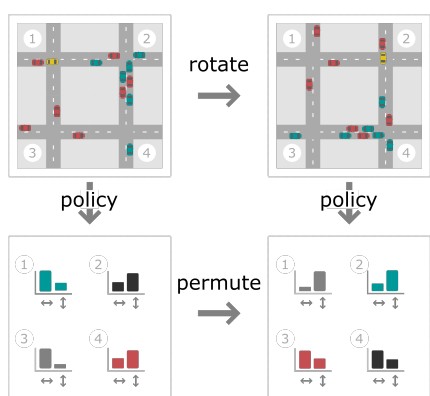

Figure 1: Example of a global multi-agent symmetry. When a rotated state occurs in the environment the optimal policy is permuted, both between and within agents.

Existing work on symmetries in single

agent reinforcement learning can only be generalized to the fully centralized multi-agent setting, because such approaches rely on the global symmetry in the full state-action spaces and these can result in correspondences *across* agents, as shown in Figure 1. Thus, such approaches cannot be used in *distributed* multi-agent systems with communication constraints. Here, we seek to be equivariant to global symmetries of cooperative multi-agent systems while still being able to execute policies in a distributed manner.

Existing work in deep multi-agent reinforcement learning has shown the potential of using permutation symmetries and invariance between agents (Liu et al., 2019; Jiang et al., 2020; Sunehag et al., 2017; Robbel et al., 2016; Böhmer et al., 2020; Sukhbaatar et al., 2016; van der Pol & Oliehoek, 2016). Such work takes an anonymity view of homogeneous agents, where the agent's observations matter for the policy but not its identity. Using permutation symmetries ensures extensive weight sharing between agents, resulting in improved data efficiency. Here, we go beyond such permutation symmetries, and consider more general symmetries of global multi-agent systems, such as rotational symmetries.

In this paper, we propose Multi-Agent MDP Homomorphic Networks, a class of distributed policy networks which are equivariant to global symmetries of the multi-agent system, as well as to standard permutation symmetries. Our contributions are as follows. (i) We propose a factorization of global symmetries in the joint state-action space of cooperative multi-agent systems. (ii) We introduce a multi-agent equivariant policy network based on this factorization. (iii) Our main contribution is an approach to cooperative multi-agent reinforcement learning that is globally equivariant while requiring only local agent computation and local communication between agents at execution time. We evaluate Multi-Agent MDP Homomorphic Networks on symmetric multi-agent problems and show improved data efficiency compared to non-equivariant baselines.

## 2 RELATED WORK

**Symmetries in single agent reinforcement learning**   Symmetries in Markov Decision Processes have been formalized by Zinkevich & Balch (2001); Ravindran & Barto (2001). Recent work on symmetries in single agent deep reinforcement learning has shown improvements in terms of data efficiency. Such work revolves around symmetries in policy networks (van der Pol et al., 2020; Simm et al., 2021), symmetric filters (Clark & Storkey, 2015), invariant data augmentation (Laskin et al., 2020; Kostrikov et al., 2021) or equivariant trajectory augmentation (Lin et al., 2020; Mavalankar, 2020; Mishra et al., 2019) These approaches are only suitable for single agent problems or centralized multi-agent controllers. Here, we solve the problem of enforcing global equivariance while still allowing distributed execution.

**Graphs and permutation symmetries in multi agent reinforcement learning**   Graph-based methods in cooperative multiagent reinforcement learning are well-explored. Much work is based around coordination graphs (Guestrin et al., 2002b;a; Kok & Vlassis, 2006), including approaches that approximate local Q-functions with neural networks and use max-plus to find a joint policy (van der Pol & Oliehoek, 2016; Böhmer et al., 2020), and approaches that use graph-structured networks to find joint policies or value functions (Jiang et al., 2020; Sukhbaatar et al., 2016). In deep learning for multi-agent systems, the use of permutation symmetries is common, either through explicit formulations (Sunehag et al., 2017; Böhmer et al., 2020) or through the use of graph or message passing networks (Liu et al., 2019; Jiang et al., 2020; Sukhbaatar et al., 2016). Policies in multi-agent systems with permutation symmetries between agents are also known as functionally homogeneous policies (Zinkevich & Balch, 2001) or policies with agent anonymity (Robbel et al., 2016; Varakantham et al., 2014). Here, we move beyond permutation symmetries to a broader group of symmetries in multiagent reinforcement learning.

**Symmetries in multi agent reinforcement learning**   Recently, Hu et al. (2020) used knowledge of symmetries to improve zero-shot coordination in games which require symmetry-breaking. Here, we instead use symmetries in cooperative multi-agent systems to improve data efficiency by parameter sharing between different transformations of the global system.

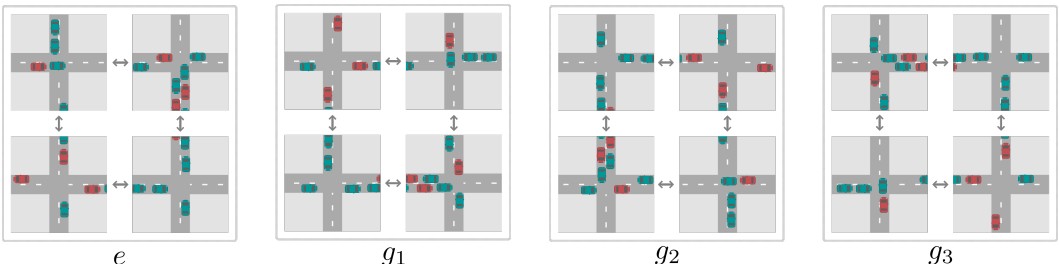

Figure 2: *The orbit of a global traffic light state under the group of 90 degree rotations.*

## 3 BACKGROUND

In this section we introduce the necessary definitions and notation used in the rest of the paper.

### 3.1 MULTI-AGENT MDPs

We will start from the definition of Multi-Agent MDPs, a class of fully observable cooperative multi-agent systems. Full observability implies that each agent can execute the same centralized policy. Later on we will define a distributed variant of this type of decision making problem.

**Definition 1** *A* Multi-Agent Markov Decision Process *(MMDP) (Boutilier, 1996) is a tuple* $(\mathcal{N}, S, \mathbf{A}, T, R)$ *where* $\mathcal{N}$ *is a set of* $m$ *agents,* $S$ *is the state space,* $\mathbf{A} = A_1 \times \cdots \times A_m$ *is the joint action space of the MMDP,* $T : S \times A_1 \times \cdots \times A_m \times S \to [0, 1]$ *is the transition function, and* $R : S \times A_1 \times \cdots \times A_m \to \mathbb{R}$ *is the immediate reward function.*

The goal of an MMDP, as in the single agent case, is to find a *joint policy* that maps states to probability distributions over joint actions, $\pi : S \to \Delta(A)$, (with $\Delta(A)$ the space of such distributions) to maximize the expected discounted return of the system, $R_t = \mathbb{E}[\sum_{k=0}^{T} \gamma^k r_{t+k+1}]$ with $\gamma \in [0, 1]$ a discount factor. An MMDP can be viewed as a single-agent MDP where the agent takes joint actions.

### 3.2 GROUPS AND TRANSFORMATIONS

In this paper we will refer extensively to group symmetries. Here, we will briefly introduce these concepts and explain their significance to discussing equivalences of decision making problems.

A group $G$ is a set with a binary operator $\cdot$ that obeys the group axioms: identity, inverse, closure, and associativity. Consider as a running example the set of 90 degree rotations $\{0°, 90°, 180°, 270°\}$, which we can write as rotation matrices:

$$R(\theta) = \begin{bmatrix} \cos\theta & -\sin\theta \\ \sin\theta & \cos\theta \end{bmatrix} \quad (1)$$

with $\theta \in \{0, \frac{\pi}{2}, \pi, \frac{3\pi}{2}\}$. Composing any two matrices in this set results in another matrix in the set, meaning the set is closed under composition. For example, composing $R(\frac{\pi}{2})$ and $R(\pi)$ results in another member of the set, in this case $R(\frac{3\pi}{2})$. Similarly, each member of the set has an inverse that is also in the set, and $R(0)$ is an identity element. Since matrix multiplication is associative, the group axioms are satisfied and the set is a group under composition.

A group action is a function $G \times X \to X$ that satisfies $ex = x$ (where $e$ is the identity element) and $(g \cdot h)x = g \cdot (hx)$. For example, the group of 90 degree rotation matrices acts on vectors to rotate them. Similar to the action of this group on vectors, we can define an action of the same group on image space: e.g., the NumPy (Harris et al., 2020) function `np.rot90` acts on the set of images. We will consider group actions on the set of states represented as image observations. We match these with group actions on policies. Since we consider discrete action spaces, a group element $g$ acting on a policy $\pi$ will be represented as a matrix multiplication of the policy with a permutation matrix.

When discussing symmetries in decision making problems, we identify sets of state-action pairs that are equivalent: if the state is transformed, the policy should be transformed as well, but potentially

with a different representation of the transformation. See Figure 1. We are interested in the case where the reward and transition functions are invariant in the orbit of state-action pairs under a symmetry group. The *orbit* of a point $v \in V$, with $V$ a vector space, is the set of all its transformations (e.g. all rotations of the point), defined as $\mathcal{O}(v) = \{gv | \forall g \in G\}$. The orbit of a point under a group forms an equivalence class. See Figure 2 for an example of an orbit of a traffic light state.

## 4 Distributing Symmetries over Multiple Agents

Consider the cooperative traffic light control system in Figure 1 that contains transformation-equivalent global state-action pairs. We first formalize global symmetries of the system similarly to symmetries in a single agent MDP. Then we will discuss how we can formulate distributed symmetries in a distributed MMDP. Finally, we introduce Multi-Agent MDP Homomorphic Networks.

### 4.1 Symmetries in MMDPs

We define symmetries in an MMDP similar to an MDP with symmetries (van der Pol et al., 2020).

**Definition 2** *An MMDP is an MMDP with symmetries if reward and transition functions are invariant under a transformation group $G$. That is, the MMDP has symmetries if there is at least one non-trivial group $G$ of transformations $L_g : S \to S$ and for every $s$, $K_g^s : A \to A$ such that*

$$R(s, a) = R(L_g[s], K_g^s[a]) \qquad \forall g \in G, s \in S, a \in A, \qquad (2)$$

$$T(s, a, s') = T(L_g[s], K_g^s[a], L_g[s']) \qquad \forall g \in G, s, s' \in S, a \in A. \qquad (3)$$

If two state-action pairs $s, a$ and $L_g[s], K_g^s[a]$ obey Eq. 2 and 3, then they are equivalent (van der Pol et al., 2020). Consider as an example the symmetries in Figure 1. These symmetries can result in correspondences across agents, for example when the observation of agent $i$ is mapped by the symmetry to another agent $j$ that is arbitrarily far away and with which there is no communication channel. In the next section, we will resolve this problem by defining distributed symmetries in terms of local observations and the communication graph defined by the state.

If we have an MMDP with symmetries, that means that there are symmetric optimal policies, i.e. if the state of the MMDP transforms, the policy transforms accordingly. The above definition of an MMDP with symmetries is only applicable to the centralized setting. If we want to be able to execute policies in a distributed manner, we will need to enforce equivariance in a distributed manner.

### 4.2 Distributed Multi-Agent Symmetries

In a distributed MMDP, agents make decisions based on local information only, i.e. the local states they observe, and the communications they receive from neighbors, defined as follows:

**Definition 3** *A Distributed Multiagent Markov Decision Process (Distributed MMDP) $(\mathcal{N}, \mathbf{S}, \mathbf{A}, T, R)$ is an MMDP where agents can communicate as specified by a graph $\mathcal{G} = (\mathcal{V}, \mathcal{E})$ with one node $v_i \in \mathcal{V}$ per agent and an edge $(i, j) \in \mathcal{E}$ if agents $i$ and $j$ can communicate. Thus, $\mathbf{S} = (\{S_i\}_{i \in \mathcal{N}}, \{E_{ij}\}_{(i,j) \in \mathcal{E}})$, with $S_i$ the set of state features observable by agent $i$, which may include shared global features, and $E_{ij}$ the set of edge features between $i$ and $j$. In a distributed MMDP, each agent's action can only depend on the local state and the communications it receives[1].*

Here, we focus on Distributed MMDPs which have a spatial component, i.e. each agent has a coordinate in some space, and the attributes of the edges between the agents in the communication graph contain spatial information as well. For instance, the attributes $e_{ij} \in E$ for edge $(i, j)$ might be the difference vector between agent $i$ and agent $j$'s coordinates. Since both agent observations and interaction edges have spatial features, a global symmetry will affect both the agent observations, the agent locations, and the features on the interaction edges. See Figure 3.

To allow a globally equivariant policy network with distributed execution, we might naively decide to restrict each agent's local policy network to be equivariant to local transformations. However, this

---

[1]Communication is constrained, i.e. agents cannot simply share their full observations with each other.

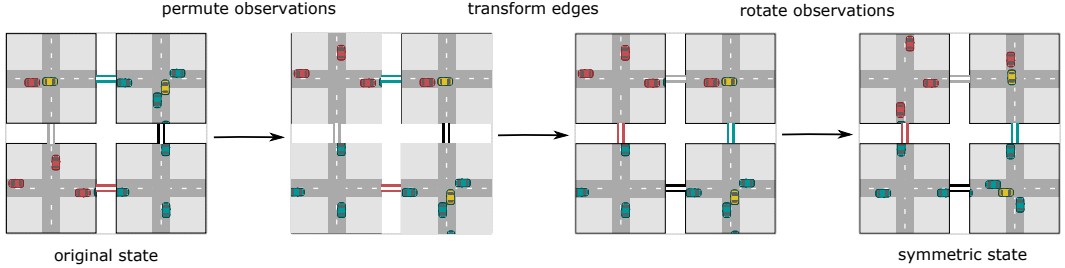

permute observations      transform edges      rotate observations

original state      symmetric state

Figure 3: *Example of how a global transformation of a distributed traffic light control state can be viewed as 1) a permutation of the observations over the agents, 2) a permutation of the interaction edges, 3) a transformation of the local observations.*

does not give us the correct global transformation, as joining the local transformations does not give us the same as the global transformations, as illustrated in Figure 4.

Instead, to get the correct transformation as shown in the left side of Figure 4, the local state is transformed, but also its position is changed, which can be seen as a permutation of the agents and their neighbors. To give an example of the equivariance constraint that we want to impose: the lower left agent (before transformation) should select an action based on its local state and communication received from its northern and eastern neighbor, while the top left agent (after transformation) should select the transformed version of the action based on its rotated local state and communication from its eastern and southern neighbor.

Since the agent has no other information about the system, if the local observations are transformed (e.g. rotated), and the messages it receives are transformed similarly, then from a local perspective the agent is in an equivalent state and should execute the same policy, but with an equivalently transformed action.

From the perspective of our agent and all its neighbors, the equivalence holds for this local subgraph as well: if the observations and local interactions rotate *relative to each other*, then the whole subgraph rotates. See Figure 3. Thus, as long as the transformations are applied to the full set of observations and the full set of communications, we have a global symmetry. We therefore propose the following definition of a Distributed MMDP with Symmetries.

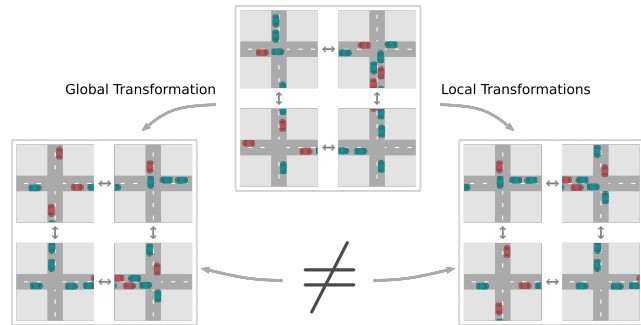

Global Transformation      Local Transformations

Figure 4: *Example of the difference between a global transformation on the global state, and a set of local transformations on local states. On the left we rotate the entire world by 90 degrees clockwise, which involves rotating crossings and streets. On the right we perform local uncoordinated transformations only at the street level. The latter is not a symmetry of the problem.*

**Definition 4** *Let* $\mathbf{s} = (\{s_i\}_{i \in \mathcal{N}}, \{e_{ij}\}_{(i,j) \in \mathcal{E}})$. *Then a* Distributed MMDP with symmetries *is a Distributed MMDP for which the following equations hold for at least one non-trivial set of group transformations* $L_g : S \to S$ *and for every* $s$, $K_g^s : A \to A$ *such that*

$$R(\mathbf{s}, \mathbf{a}) = R(L_g[\mathbf{s}], K_g^{\mathbf{s}}[\mathbf{a}]) \qquad\qquad \forall g \in G, \mathbf{s} \in \mathbf{S}, \mathbf{a} \in \mathbf{A} \qquad (4)$$

$$T(\mathbf{s}, \mathbf{a}, \mathbf{s}') = T(L_g[\mathbf{s}], K_g^{\mathbf{s}}[\mathbf{a}], L_g[\mathbf{s}']) \qquad\qquad \forall g \in G, \mathbf{s}, \mathbf{s}' \in \mathbf{S}, \mathbf{a} \in \mathbf{A} \qquad (5)$$

*where equivalently to acting on* $\mathbf{s}$ *with* $L_g$, *we can act on the interaction and agent features separately with* $\tilde{L}_g$ *and* $U_g$, *to end up in the same global state:*

$$L_g[\mathbf{s}] = (P_g^s[\{\tilde{L}_g[s_i]\}_{i \in \mathcal{N}}], P_g^e[\{U_g[e_{ij}]\}_{(i,j) \in \mathcal{E}}]) \qquad (6)$$

*for* $\tilde{L}_g : S_i \to S_i$, $U_g : E \to E$, *and* $P_g^s$ *and* $P_g^e$ *the state and edge permutations.*

Here, $E$ is the set of edge features. The symmetries acting on the agents and agent interactions in the Distributed MMDP are a class of symmetries we call *distributable symmetries*. We have now defined a class of Distributed MMDPs with symmetries for which we can distribute a global symmetry into a set of symmetries on agents and agent interactions. This distribution allows us to define distributed policies that respect the global symmetry.

### 4.3 MULTI-AGENT MDP HOMOMORPHIC NETWORKS

We have shown above how distributable global symmetries can be decomposed into local symmetries on agent observations and agent interactions. Here, we discuss how to implement distributable symmetries in multi-agent systems in practice.

#### 4.3.1 GENERAL FORMULATION

We want to build a neural network that 1) allows distributed execution, so that we can compute policies without a centralized controller 2) allows us to pass communications between agents (agent interactions), to enable coordination and 3) exhibits the following global equivariance constraint:

$$\vec{\pi}_\theta(L_g[\mathbf{s}]) = H_g^s[\vec{\pi}_\theta(\mathbf{s})] \tag{7}$$

with $H_g^s : \Pi \to \Pi$. Thus, the policy network $\vec{\pi}$ that outputs the joint policy must be *equivariant* under group transformations of the global state. To satisfy 1) and 2), i.e. allowing distributed execution and agent-to-agent communication, as well as permutation equivariance, we formulate the network as a message passing network (MPN), but with global equivariance constraints.

$$H_g^s[\vec{\pi}] = \text{MPN}_\theta(L_g[s]) \tag{8}$$

Since a network is end-to-end equivariant if all its layers are equivariant with matching representations (Cohen & Welling, 2016), we require layer-wise equivariance constraints on the layers. A single layer in an MPN is given by a set of node updates, i.e. $f_i^{(l+1)} = \phi_u\left(f_i^{(l)}, \sum_{j=1}^{|\mathcal{N}_i|} \phi_m\left(e_{ij}, f_j^{(l)}\right)\right)$, with $f_j^{(l)}$ the current encoding of agent $j$ in layer $l$, $\phi_m$ the message function that computes $m_{j \to i}$ based on the edge $e_{ij}$ and the current encoding of agent $j$, and $\phi_u$ the node update function that updates agent $i$'s current encoding based on $f_i^{(l)}$ and the aggregated received message $m_i^{(l)}$. Since the layer is given by a set of node updates, the equivariance constraint is on the node updates. In other words, $\phi_u$ must be an equivariant function of local encoding $f_i^{(l)}$ and aggregated message $m_i^{(l)}$ :

$$P_g\left[\phi_u(f_i^{(l)}, m_i^{(l)})\right] = \phi_u\left(L_g[f_i^{(l)}], L_g[m_i^{(l)}]\right) \tag{9}$$

Thus, the node update function $\phi_u$ is constrained to be equivariant to transformations of inputs $f_i^{(l)}$ and $m_i^{(l)}$. Therefore, to conclude that the outputs of $\phi_u$ transform according to $P_g$, we only need to enforce that its inputs $f_i$ and $m_i$ transform according to $L_g$. Thus, the subfunction $\phi_m$ that computes the messages $m_i^{(l)}$ must be constrained to be equivariant as well. Since $\phi_m$ takes input the previous layer's encodings as well as the edges $e_{ij}$, this means that 1) the encodings must contain geometric information about the state, e.g. which rotation the local state is in and 2) the edge attributes contain geometric information as well, i.e. they transform when the global state transforms (Appendix B).

$$L_g\left[m_i^{(l)}\right] = \sum_{j=1}^{|\mathcal{N}_i|} \phi_m\left(U_g\left[e_{ij}\right], L_g[f_j^{(l)}]\right) \tag{10}$$

Note that this constraint is satisfied when $\phi_m$ is equivariant, since linear combinations of equivariant functions are also equivariant (Cohen & Welling, 2017). Putting this all together, the local encoding $f_i^{(l)}$ for each agent is equivariant to the set of edge rotations and the set of rotations of encodings in the previous layer. For more details, see Appendix B. Thus, we now have the general formulation of Multi-Agent MDP Homomorphic Networks. At execution time, the distributed nature of Multi-Agent MDP Homomorphic Networks allows them to be copied onto different devices and messages exchanged between agents only locally, while still enforcing the global symmetries.

### 4.3.2 MULTI-AGENT MDP HOMOMORPHIC NETWORK ARCHITECTURE

Multi-Agent MDP Homomorphic Networks consist of equivariant local observation encoders $\phi_e$ : $S_i \to \mathbb{R}^{|G| \times D}$, where $G$ is the group, $|G|$ is its size, and $D$ the dimension of the encoding, equivariant local message functions $\phi_m : \mathcal{E} \times \mathbb{R}^{|G| \times D} \to \mathbb{R}^{|G| \times F}$ where $F$ is dimension of the message encoding, equivariant local update functions $\phi_u : \mathbb{R}^{|G| \times D} \times \mathbb{R}^{|G| \times F} \to \mathbb{R}^{|G| \times D}$, and equivariant local policy predictors $\phi_\pi : \mathbb{R}^{|G| \times D} \to \Pi(A_i)$. Take the example of multi-agent traffic light control with 90 degree rotation symmetries, which we evaluate on in Section 5. In this setting, we wish to constrain $\phi_e$ to be equivariant to rotations $R_g$ of the local observations. We will require the outputs of $\phi_e$ to permute according to $L_g^{-1}$ whenever the input rotates by $R_g$.

$$L_g\left[\phi_e(s_i)\right] = \phi_e(R_g\left[s_i\right]) \quad \forall g \in G \tag{11}$$

This has the form of a standard equivariance constraint, which allows conventional approaches to enforcing group equivariance, e.g. (Cohen & Welling, 2016). In this paper, we will enforce group equivariance using the Symmetrizer (van der Pol et al., 2020). Before training, the Symmetrizer enforces group (e.g. rotational) symmetries by projecting neural network weights onto an equivariant subspace, and then uses SVD to find a basis for the equivariant subspace. Then, during and after training, the weight matrix of the neural network is realised as a linear combination of equivariant basis weights, and the coefficients of the linear combination are updated during training with PPO (Schulman et al., 2017). We use ReLU non-linearities and regular representations. For more details on the Symmetrizer, we refer the reader to van der Pol et al. (2020).

After encoding the input observations with the equivariant encoding function $\phi_e$, we have an equivariant encoding of the local states that has a compositional form: the rotation of the state is represented by the ordering of *group channels* (see van der Pol et al. (2020)) and the other state features are represented by the information in those channels. Similarly, we constrain the message update functions to be equivariant to the permutation $L_g$ in the group channels of the state encodings and the rotation $U_g$ of a difference vector $e_{ij}$, representing the edge $(i, j)$:

$$L_g\left[\phi_m\left(e_{ij}, f_j^{(l)}\right)\right] = \phi_m\left(U_g\left[e_{ij}\right], L_g\left[f_j^{(l)}\right]\right) \tag{12}$$

Since linear combinations of equivariant functions are equivariant as well (Cohen & Welling, 2017), the aggregated message $m_i^{(l)} = \sum_j^{|\mathcal{N}_i|} m_{j \to i}$ is equivariant too.

While $e_{ij}$ and $f_j^{(l)}$ transform under the same group $G$, they do not transform under the same group *action*: $e_{ij}$ is a vector that transforms with a rotation matrix, whereas $f_j^{(l)}$ transforms with a permutation of group channels. The question arises how to build group equivariant layers that transform both the edge and the agent features appropriately. The method we use is to build equivariant layers using direct sum representations, where the representations $U_g$ and $L_g$ are combined as follows:

$$T_g = U_g \oplus L_g = \begin{bmatrix} U_g & 0 \\ 0 & L_g \end{bmatrix} \tag{13}$$

where $0$ represents a zero-matrix of the appropriate size. Consider a weight matrix $W^l$ acting on $\begin{bmatrix} e_{ij} \\ f_j^{(l)} \end{bmatrix}$. The equivariance constraint then becomes $W^{(l)} T_g = L_g W^{(l)}$.

To preserve the geometric information coming from the messages, the node update function is similarly constrained to be equivariant. Importantly, the permutation on the outputs of $\phi_u$ must match the permutation on the inputs of the next layer's $\phi_m$ (i.e. the output of one layer must use the same group representation as the input of the next layer). This ensures that we can add multiple layers together while preserving the geometric information. In practice, it is convenient to use a single representation $L_g$ for all permutation representations.

$$L_g\left[\phi_u(f_i^{(l)}, m_i^{(l)})\right] = \phi_u\left(L_g[f_i^{(l)}], L_g[m_i^{(l)}]\right) \tag{14}$$

Finally, after $M$ layers ($M$ message passing rounds), we output local equivariant policies based on the state encodings at layer $M$ using local policy network $\pi$:

$$\pi_i\left(L_g\left[f_i^{(M)}\right]\right) = P_g\left[\pi_i\left(f_i^{(M)}\right)\right] \tag{15}$$

Here, $P_g$ is the permutation representation on the actions of the individual agent, e.g. if in a grid world the state is flipped, $P_g$ is the matrix that permutes the left and right actions accordingly.

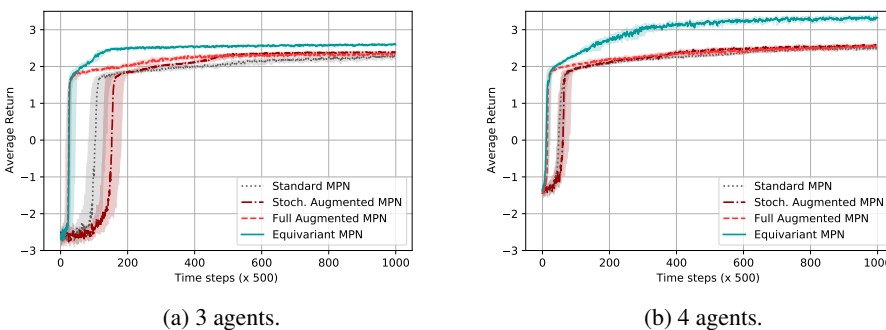

(a) 3 agents.                    (b) 4 agents.

Figure 5: *Results for the distributed drone wildlife monitoring task. 25%, 50% and 75% quantiles shown over 15 random seeds. All approaches tuned over 6 learning rates.*

## 5 EXPERIMENTS

The evaluation of Multi Agent MDP Homomorphic networks has a singular goal: to investigate and quantify the effect of distributed versions of global equivariance in symmetric cooperative multi-agent reinforcement learning. We compare to three baselines. The first is a non-homomorphic variant of our network. This is a standard MPN, which is a permutation equivariant multi-agent graph network but not equivariant to global rotations. The other two are variants with symmetric data augmentation, in the spirit of (Laskin et al., 2020; Kostrikov et al., 2021). For a stochastic data augmentation baseline, on each forward pass one of the group elements is sampled and used to transform the input, and appropriately transform the output as well. For a full augmentation baseline, every state and policy is augmented with all its rotations in the group. For evaluation, we use the common centralized training, decentralized execution paradigm (Kraemer & Banerjee, 2016; Oliehoek et al., 2008) (see Appendix A for more details). We train in a centralized fashion, with PPO (Schulman et al., 2017), which will adjust the coefficients of the weight matrices in the network. The information available to the actors and critics is their local information and the information received from neighbors. We first evaluate on a wildlife monitoring task, a variant of predator-prey type problems with pixel-based inputs where agents can have information none of the other agents have. Additionally, we evaluate the networks on the more complex coordination problem of traffic light control, with pixel-based inputs. We focus on C4 as the discrete group to investigate whether equivariance improves multi-agent systems, as C4 has been shown to be effective in supervised learning and single-agent settings.

### 5.1 WILDLIFE MONITORING

**Setup**   We evaluate on a distributed wildlife monitoring setup, where a set of drones has to coordinate to trap poachers. To trap a poacher, one drone has to hover above them while the other assists from the side, and for each drone that assists the team receives +1 reward. Two drones cannot be in the same location at the same time. Since the drones have only cameras mounted at the bottom, they cannot see each other. The episode ends when the poacher is trapped by at least 2 drones, or 100 time steps have passed. On each time step the team gets -0.05 reward. All agents (including the poacher) can stand still or move in the compass directions. The poacher samples actions uniformly. We train for 500k time steps. The drones can send communications to drones within a 3 by 3 radius around their current location, meaning that the problem is a distributed MMDP. Due to changing agent locations and the limited communication radius, the communication graph is dynamic and can change between time steps. The observations are 21 by 21 images representing a agent-centric view of a 7 by 7 toroidal grid environment that shows where the target is relative to the drone. While the grid is toroidal, the communication distance is not: at the edges of the grid, communication is blocked. This problem exhibits 90 degree rotations: when the global state rotates, the agents' local policies should permute, and so should the probabilities assigned to the actions in the local policies.

**Results**   Results for this task are shown in Figure 5, with on the y-axis the average return and on the x-axis the number of time steps. In both the 3-agent and 4-agent case, using a Multi-Agent

MDP Homomorphic Network improves compared to using MPNs without symmetry information, and compared to using symmetric data augmentation. We conclude that in the proposed task, our approach learns effective joint policies in fewer environment interactions compared to the baselines.

## 5.2 TRAFFIC LIGHT CONTROL

For a second experiment, we focus on a more complex coordination problem: reducing vehicle wait times in traffic light control. Traffic light control constitutes a longstanding and open problem (see Wei et al. (2019) for an overview): not only is the optimal coordination strategy non-obvious, traffic light control is a problem where wrong decisions can quickly lead to highly suboptimal states due to congestion. We use this setting to answer the following question: does enforcing symmetries help in complex coordination problems?

**Setup** We use a traffic simulator with four traffic lights. On each of eight entry roads, for 100 time steps, a vehicle enters the simulation on each step with probability 0.1. Each agent controls the lights of a single intersection and has a local action space of (`grgr`, `rgrg`), indicating which two of its four lanes get a red or green light. Vehicles move at a rate of one unit per step, unless they are blocked by a red light or a vehicle. If blocked, the vehicle needs one step to restart. The goal is reducing the average vehicle waiting time. The simulation ends after all vehicles have exited the system, or after 500 steps. The team reward is $-\frac{1}{1000}\frac{1}{C}\sum_{c\in C} w(c)$, with $C$ the vehicles in the system and $w(c)$ vehicle $c$'s cumulative waiting time.

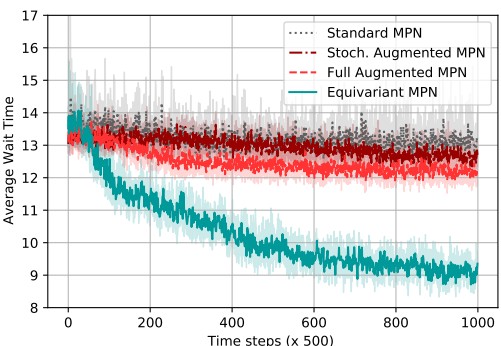

Figure 6: *Average vehicle wait times for distributed settings of the traffic light control task. Graphs show 25%, 50% and 75% quantiles over 20 independent training runs. All approaches tuned over 6 learning rates.*

**Results** We show results in Figure 6. While the standard MPN architecture had reasonable performance on the toy problem, it takes many environment interactions to improve the policy in the more complex coordination problem presented by traffic light control. Adding data augmentation helps slightly. However, we see that enforcing the global symmetry helps the network find an effective policy much faster. In this setting, the coordination problem is hard to solve: in experiments with centralized controllers, the standard baseline performs better, though it is still slower to converge than the equivariant centralized controller. Overall, enforcing global symmetries in distributed traffic light control leads to effective policies in fewer environment interactions.

## 6 CONCLUSION

We consider distributed cooperative multi-agent systems that exhibit global symmetries. In particular, we propose a factorization of global symmetries into symmetries on local observations and local interactions. On this basis, we propose Multi-Agent MDP Homomorphic Networks, a class of policy networks that allows distributed execution while being equivariant to global symmetries. We compare to non-equivariant distributed networks, and show that global equivariance improves data efficiency on both a predator-prey variant, and on the complex coordination problem of traffic light control.

**Scope** We focus on discrete groups. For future work, this could be generalized by using steerable representations, at the cost of not being able to use pointwise nonlinearities. We also focus on discrete actions. This might be generalized by going beyond action permutations, e.g., for 2D continuous worlds a continuous rotation of the actions. Furthermore, our approach uses group channels for each layer (regular representations). For small groups this is not an issue, but for much larger groups this would require infeasible computational resources. Finally, this work has focused on exact symmetries and considers imperfect symmetries and violations of symmetry constraints a promising future topic.

## 7 ACKNOWLEDGMENTS

We thank Ian Gemp, Pascal Mettes, and Patrick Forré for helpful comments.

F.A.O. received funding from the European Research Council (ERC) under the European Union's Horizon 2020 research and innovation programme (grant agreement No. 758824 —INFLUENCE).

## 8 ETHICS STATEMENT

Our work has several potential future applications, e.g. in autonomous driving, decentralized smart grids or robotics. Such applications hopefully have a positive societal impact, but there are also risks of negative societal impact: through the application itself (e.g. military), labor market impact, or by use in safety-critical applications without proper verification and validation. These factors should be taken into account when developing such applications.

## 9 REPRODUCIBILITY STATEMENT

To ensure reproducibility, we describe our setup in the Experiments section and we include hyperparameters, group actions, and architecture details in the Appendix. Our code is available at `https://github.com/ElisevanderPol/marl_homomorphic_networks`.

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

## A  MESSAGE PASSING NETWORKS, COMMUNICATION, AND DISTRIBUTION

Message passing algorithms are commonly used to coordinate between agents while allowing for factorization of the global decision making function (Guestrin et al., 2002b; Kuyer et al., 2008; van der Pol & Oliehoek, 2016; Böhmer et al., 2020). Message passing networks approximate such message passing algorithms (Yoon et al., 2019; Satorras & Welling, 2021). Thus, we can view message passing networks as a type of learned communication between coordinating agents. Message passing networks can be executed using only local communication and computation. To see that this is the case, consider the equations that determine the message passing networks in this paper:

$$f_i^{(0)} = \phi_e(s_i) \tag{16}$$

$$m_{j\to i}^{(l)} = \phi_m(e_{ij}, f_j^{(l)}) \tag{17}$$

$$m_i^{(l)} = \sum_{j}^{|\mathcal{N}_i|} m_{j\to i} \tag{18}$$

$$f_i^{(l+1)} = \phi_u(f_i^{(l)}, m_i^{(l)}) \tag{19}$$

for all agents $i$ and layers (message passing rounds) $l$. In Eq. 16, each agent encodes its local observation $s_i$ into a local feature vector $f_i^{(0)}$. In Eq. 17, each agent $j$ computes its message to agent $i$ using its own local feature vector $f_j^{(l)}$ and its shared edge features $e_{ij} = x_i - x_j$. After agent $i$ receives its neighbors' messages, it aggregates them in Eq. 18. Finally, in Eq. 19 agent $i$ updates its local feature vector using the aggregated message and its local feature vector. Clearly, every step in this network requires only local computation and local communication, therefore allowing the network to be distributed over agents at execution time.

## B  EQUIVARIANCE OF PROPOSED MESSAGE PASSING LAYERS

Recall that $e_{ij} = x_i - x_j$ (the difference between the locations of agent $i$ and agent $j$). Therefore,

$$U_g[e_{ij}] = U_g[x_i - x_j] = U_g[x_i] - U_g[x_j] \tag{20}$$

So, when both agent $i$ and agent $j$ are moved to a location transformed by $U_g$, the edge features $e_{ij}$ are transformed by $U_g$ as well. If all agent positions and observations rotate by the same underlying group, this means the full system has rotated. In this paper, we place equivariance constraints on the message function:

$$K_g[\sum_{j} \phi_m(e_{ij}, f_j)] = \sum_{j} \phi_m(U_g[e_{ij}], L_g[f_j]) \tag{21}$$

This means that the messages are only permuted by $K_g$ if both the local features $f_j$ *and* the edge features $e_{ij}$ are transformed (by $L_g$ and $U_g$ respectively). To see that the proposed message passing layers are equivariant to transformations on agent features and edge features, consider the following example. Assume we have a message passing layer $\phi$ consisting of node update function $\phi_u$ and message update function $\phi_m$, such that with agent features $\{f_i\}$, and edge features $\{e_{ij}\}$, the output for agent $i$ is given by

$$\phi^{(i)}(\{e_{ij}\}, \{f_j\}) = \phi_u(f_i, m_i) \tag{22}$$

$$= \phi_u\left(f_i, \sum_{j} \phi_m(e_{ij}, f_j)\right) \tag{23}$$

Assume $\phi_u$ and $\phi_m$ are constrained to be equivariant in the following way:

$$P_g[\phi_u(f_i, m_i)] = \phi_u(L_g[f_i], K_g[m_i]) \tag{24}$$

$$K_g[\sum_{j} \phi_m(e_{ij}, f_j)] = \sum_{j} \phi_m(U_g[e_{ij}], L_g[f_j]) \tag{25}$$

for group elements $g \in G$. Then, for layer $\phi$:

$$P_g \left[ \phi^{(i)} \left( \{e_{ij}\}, \{f_j\} \right) \right] = P_g \left[ \phi_u \left( f_i, m_i \right) \right] \tag{26}$$

$$= P_g \left[ \phi_u \left( f_i, \sum_j \phi_m \left( e_{ij}, f_j \right) \right) \right] \tag{27}$$

$$= \phi_u \left( L_g \left[ f_i \right], K_g \left[ \sum_j \phi_m \left( e_{ij}, f_j \right) \right] \right) \quad \text{(using Eq. 24)} \tag{28}$$

$$= \phi_u \left( L_g \left[ f_i \right], \sum_j \phi_m \left( U_g \left[ e_{ij} \right], L_g \left[ f_j \right] \right) \right) \quad \text{(using Eq. 25)} \tag{29}$$

$$= \phi^{(i)} \left( \{U_g \left[ e_{ij} \right]\}, \{L_g \left[ f_j \right]\} \right) \tag{30}$$

## C   DISCRETE ROTATIONS OF CONTINUOUS VECTORS

Here we outline how to build weight matrices equivariant to discrete rotations of continuous vectors. Let $e_{ij} = x_j - x_i$ be an arbitrary, continuous difference vector between the coordinates of agent $j$ and the coordinates of agent $i$. Then this difference vector transforms under 90 degree rotations using the group of standard 2D rotation matrices of the form

$$R_g = R(\theta) = \begin{bmatrix} \cos\theta & -\sin\theta \\ \sin\theta & \cos\theta \end{bmatrix} \tag{31}$$

for $\theta \in [0, \frac{\pi}{2}, \pi, \frac{3\pi}{2}]$. A weight matrix $W$ is now equivariant to 90 degree rotations of $e_{ij}$ if

$$K_g W e_{ij} = W R_g e_{ij} \tag{32}$$

with $\{K_g\}_{g \in G}$ e.g. a permutation matrix representation of the same group. So,

$$W = K_g^{-1} W R_g \tag{33}$$

which we can solve using standard approaches.

## D   EXPERIMENTAL DETAILS

For all approaches, including baselines, we run at least 15 random seeds for 6 different learning rates, $\{0.001, 0.003, 0.0001, 0.0003, 0.00001, 0.00003\}$, and report the best learning rate for each. Other hyperparameters are taken as default in the codebase (Stooke & Abbeel, 2019; van der Pol et al., 2020).

### D.1   LEARNING RATES REPORTED

After tuning the learning rate, we report the best one for each approach. See Table 1.

| Distributed Settings | Standard MPN | Augmented MPN | Equivariant MPN |
|---|---|---|---|
| Drones, 3 agents | 0.001 | 0.0003 | 0.001 |
| Drones, 4 agents | 0.0003 | 0.001 | 0.001 |
| Traffic, 4 agents | 0.0001 | 0.0001 | 0.0001 |

Table 1: Best learning rates for distributed settings

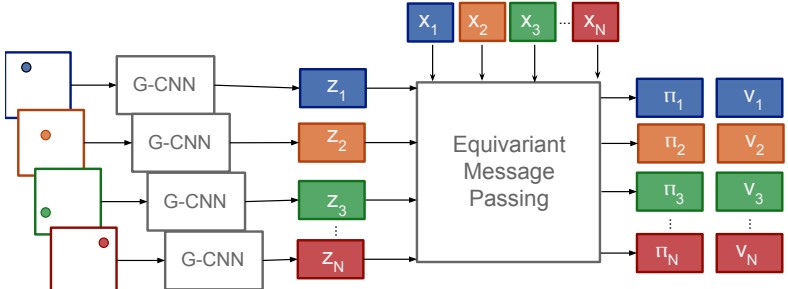

Figure 7: General overview of Multi-Agent MDP Homomorphic Networks. G-CNN refers to a group-equivariant CNN encoder. Equivariant message passing refers to the proposed equivariant message passing networks. Encoding local states with group-CNNs ensure the state encodings $z_i$ are group-equivariant. The locations $x_i$ are input to the equivariant message passing network.

## D.2 ASSETS USED

- Numpy (Harris et al., 2020) 1.19.2: BSD 3-Clause "New" or "Revised" License;
- PyTorch (Paszke et al., 2017) 1.2.0: Modified BSD license;
- RLPYT (Stooke & Abbeel, 2019): MIT license;
- MDP Homomorphic Networks & Symmetrizer (van der Pol et al., 2020): MIT license.

## E ARCHITECTURAL DETAILS

Architectures are given below and were chosen to be as similar as possible between different approaches, keeping the number of trainable parameters comparable between approaches. We chose 2 message passing layers to allow for 2 message passing hops. For the message passing networks, we use L1-normalization of the adjacency matrix.

### E.1 ARCHITECTURAL OVERVIEW

The global structure of our network is given in Figure 7.

### E.2 ARCHITECTURES

#### E.2.1 WILDLIFE MONITORING

Listing 1: Equivariant Network Architecture for Centralized Drones

```
EqConv2d(repr_in=1, channels_in=m, repr_out=4, channels_out=⌊16/√4⌋,
           filter_size=(7, 7), stride=2, padding=0)
ReLU()
EqConv2d(repr_in=4, channels_in=⌊16/√4⌋, repr_out=4, channels_out=⌊32/√4⌋,
           filter_size=(5, 5), stride=1, padding=0)
ReLU()
GlobalMaxPool()
EqLinear(repr_in=4, channels_in=⌊32/√4⌋, repr_out=4, channels_out=⌊128/√4⌋)
ReLU()
EqLinear(repr_in=4, channels_in=⌊128/√4⌋, repr_out=5, channels_out=⌊64/√4⌋)
ReLU()
ModuleList([EqLinear(repr_in=4, channels_in=⌊64/√4⌋, repr_out=5,
  channels_out=1) for i in range(m)])
EqLinear(repr_in=4, channels_in=⌊64/√4⌋, repr_out=1, channels_out=1)
```

Listing 2: CNN Architecture for Centralized Drones

```
1  Conv2d(channels_in=m, channels_out=16,
2          filter_size=(7, 7), stride=2, padding=0)
3  ReLU()
4  Conv2d(channels_in=16,channels_out=32,
5          filter_size=(5, 5), stride=1, padding=0)
6  ReLU()
7  GlobalMaxPool()
8  Linear(channels_in=32, channels_out=256)
9  ReLU()
10 ModuleList([Linear(channels_in=256,
11   channels_out=5) for i in range(m)])
12 Linear(channels_in=256, channels_out=1)
```

Listing 3: Equivariant Network Architecture for Distributed Drones

```
1  EqConv2d(repr_in=1, channels_in=1, repr_out=4, channels_out=⌊16/√4⌋,
2          filter_size=(7, 7), stride=2, padding=0)
3  ReLU()
4  EqConv2d(repr_in=4, channels_in=⌊16/√4⌋, repr_out=4, channels_out=⌊32/√4⌋,
5          filter_size=(5, 5), stride=1, padding=0)
6  ReLU()
7  GlobalMaxPool()
8  EqMessagePassingLayer(repr_in=4+4+2, channels_in=⌊32/√4⌋, repr_out=4,
9    channels_out=⌊64/√4⌋)
10 ReLU()
11 EqMessagePassingLayer(repr_in=4+4+2, channels_in=⌊64/√4⌋, repr_out=4,
12   channels_out=⌊64/√4⌋)
13 ReLU()
14 EqMessagePassingLayer(repr_in=4, channels_in=⌊64/√4⌋, repr_out=5,
15   channels_out=1)
16 EqMessagePassingLayer(repr_in=4, channels_in=⌊64/√4⌋, repr_out=1,
17   channels_out=1)
```

Listing 4: MPN Architecture for Distributed Drones

```
1  Conv2d(channels_in=1, channels_out=16,
2          filter_size=(7, 7), stride=2, padding=0)
3  ReLU()
4  Conv2d(channels_in=16, channels_out=32,
5          filter_size=(5, 5), stride=1, padding=0)
6  ReLU()
7  GlobalMaxPool()
8  MessagePassingLayer(channels_in=32+32+2, channels_out=64)
9  ReLU()
10 MessagePassingLayer(channels_in=64+64+2, channels_out=64)
11 ReLU()
12 Linear(channels_in=64, channels_out=5)
13 Linear(channels_in=64, channels_out=1)
```

### E.2.2   TRAFFIC LIGHT CONTROL

Listing 5: Equivariant Network Architecture for Centralized Traffic

```
1  EqConv2d(repr_in=1, channels_in=3, repr_out=4, channels_out=⌊16/√4⌋,
2          filter_size=(7, 7), stride=2, padding=0)
3  ReLU()
4  EqConv2d(repr_in=4, channels_in=⌊16/√4⌋, repr_out=4, channels_out=⌊32/√4⌋,
5          filter_size=(5, 5), stride=1, padding=0)
6  ReLU()
```

```
 7  GlobalMaxPool()
 8  EqLinear(repr_in=4, channels_in=⌊32/√4⌋, repr_out=4, channels_out=⌊128/√4⌋)
 9  ReLU()
10  EqLinear(repr_in=4, channels_in=⌊128/√4⌋, repr_out=5, channels_out=⌊64/√4⌋)
11  ReLU()
12  EqLinear(repr_in=4, channels_in=⌊64/√4⌋, repr_out=8, channels_out=1)
13  EqLinear(repr_in=4, channels_in=⌊64/√4⌋, repr_out=1, channels_out=1)
```

### Listing 6: CNN Architecture for Centralized Traffic

```
 1  Conv2d(channels_in=3, channels_out=16,
 2              filter_size=(7, 7), stride=2, padding=0)
 3  ReLU()
 4  Conv2d(channels_in=16,channels_out=32,
 5              filter_size=(5, 5), stride=1, padding=0)
 6  ReLU()
 7  GlobalMaxPool()
 8  Linear(channels_in=32, channels_out=256)
 9  ReLU()
10  Linear(channels_in=256, channels_out=8)
11  Linear(channels_in=256, channels_out=1)
```

### Listing 7: Equivariant Network Architecture for Distributed Traffic

```
 1  EqConv2d(repr_in=1, channels_in=3, repr_out=4, channels_out=⌊16/√4⌋,
 2              filter_size=(7, 7), stride=2, padding=0)
 3  ReLU()
 4  EqConv2d(repr_in=4, channels_in=⌊16/√4⌋, repr_out=4, channels_out=⌊32/√4⌋,
 5              filter_size=(5, 5), stride=1, padding=0)
 6  ReLU()
 7  GlobalMaxPool()
 8  EqMessagePassingLayer(repr_in=4+4+2, channels_in=⌊32/√4⌋, repr_out=4,
 9    channels_out=⌊64/√4⌋)
10  ReLU()
11  EqMessagePassingLayer(repr_in=4+4+2, channels_in=⌊64/√4⌋, repr_out=4,
12    channels_out=⌊64/√4⌋)
13  ReLU()
14  EqMessagePassingLayer(repr_in=4, channels_in=⌊64/√4⌋, repr_out=2,
15    channels_out=1)
16  EqMessagePassingLayer(repr_in=4, channels_in=⌊64/√4⌋, repr_out=1,
17    channels_out=1)
```

### Listing 8: MPN Architecture for Distributed Traffic

```
 1  Conv2d(channels_in=3, channels_out=16,
 2              filter_size=(7, 7), stride=2, padding=0)
 3  ReLU()
 4  Conv2d(channels_in=16, channels_out=32,
 5              filter_size=(5, 5), stride=1, padding=0)
 6  ReLU()
 7  GlobalMaxPool()
 8  MessagePassingLayer(channels_in=32+32+2, channels_out=64)
 9  ReLU()
10  MessagePassingLayer(channels_in=64+64+2, channels_out=64)
11  ReLU()
12  Linear(channels_in=64, channels_out=2)
13  Linear(channels_in=64, channels_out=1)
```

### E.3 GROUP ACTIONS

Here we list the group actions used in different equivariant layers throughout our experiments. For all equivariant layers, we use the Symmetrizer (van der Pol et al., 2020) to find equivariant weight bases.

**Rotation-equivariant Filters**  For all equivariant encoder networks, we create 90 degree rotation-equivariant filters using `np.rot90`.

### E.3.1 GROUP ACTIONS FOR WILDLIFE MONITORING

**Linear layers**  Permutation matrices representing the following permutations:
$e = [0, 1, 2, 3]$
$g_1 = [3, 0, 1, 2]$
$g_2 = [2, 3, 0, 1]$
$g_3 = [1, 2, 3, 0]$

**Policy layers, centralized**  Permutation matrices representing the following permutations:
$e = [0, 1, 2, 3, 4]$
$g_1 = [0, 2, 3, 4, 1]$
$g_2 = [0, 3, 4, 1, 2]$
$g_3 = [0, 4, 1, 2, 3]$

**Value layers, centralized**  Permutation matrices representing the following permutations:
$e = [1]$
$g_1 = [1]$
$g_2 = [1]$
$g_3 = [1]$

**Message Passing Layers**  Acting on state features, permutation matrices representing the following permutations:
$e = [0, 1, 2, 3]$
$g_1 = [3, 0, 1, 2]$
$g_2 = [2, 3, 0, 1]$
$g_3 = [1, 2, 3, 0]$
Acting on edge features, the following rotation matrices:
```
e=np.eye(2)
g1=np.array([[0, -1], [1, 0]])
g2=np.array([[-1, 0], [0, -1]])
g3=np.array([[0, 1], [-1, 0]])
```

**Policy layers, distributed**  Permutation matrices representing the following permutations:
$e = [0, 1, 2, 3, 4]$
$g_1 = [0, 2, 3, 4, 1]$
$g_2 = [0, 3, 4, 1, 2]$
$g_3 = [0, 4, 1, 2, 3]$

**Value layers, distributed**  Permutation matrices representing the following permutations:
$e = [1]$
$g_1 = [1]$
$g_2 = [1]$
$g_3 = [1]$

### E.3.2  GROUP ACTIONS FOR TRAFFIC LIGHT CONTROL

**Linear layers**    Permutation matrices representing the following permutations:
$e = [0, 1, 2, 3]$
$g_1 = [3, 0, 1, 2]$
$g_2 = [2, 3, 0, 1]$
$g_3 = [1, 2, 3, 0]$

**Policy layers, centralized**    Permutation matrices representing the following permutations:
$e = [0, 1, 2, 3, 4, 5, 6, 7]$
$g_1 = [5, 4, 1, 0, 7, 6, 3, 2]$
$g_2 = [6, 7, 4, 5, 2, 3, 0, 1]$
$g_3 = [3, 2, 7, 6, 1, 0, 5, 4]$

**Value layers, centralized**    Permutation matrices representing the following permutations:
$e = [1]$
$g_1 = [1]$
$g_2 = [1]$
$g_3 = [1]$

**Message Passing Layers**    Acting on state features, permutation matrices representing the following permutations:
$e = [0, 1, 2, 3]$
$g_1 = [3, 0, 1, 2]$
$g_2 = [2, 3, 0, 1]$
$g_3 = [1, 2, 3, 0]$
Acting on edge features, the following rotation matrices:
```
e =np.eye(2)
g_1 =np.array([[0, -1], [1, 0]])
g_2 =np.array([[-1, 0], [0, -1]])
g_3 =np.array([[0, 1], [-1, 0]])
```

**Policy layers, distributed**    Permutation matrices representing the following permutations:
$e = [0, 1]$
$g_1 = [1, 0]$
$g_2 = [0, 1]$
$g_3 = [1, 0]$

**Value layers, distributed**    Permutation matrices representing the following permutations:
$e = [1]$
$g_1 = [1]$
$g_2 = [1]$
$g_3 = [1]$

