# OpenReview forum: "Multi-Agent MDP Homomorphic Networks"
_ICLR.cc/2022/Conference — ICLR 2022 Poster_

### Official Review · Reviewer_xEvs · 2021-10-31

**Correctness:** 3
**Technical Novelty And Significance:** 4
**Empirical Novelty And Significance:** 3
**Recommendation:** 5
**Confidence:** 4

**Main Review:**

Strengths:

1- The paper is clearly written and easy to follow.

2- The illustrative examples in figure 1 and 4 are helpful.

3- The related work is comprehensive.

Weaknesses:

1- The authors gave a detailed analysis of how to achieve distributed symmetries in Sec 4. However, how to perform the training of policies is not given. Though it is briefly mentioned that the authors used CTDE (centralized training and decentralized execution), I would appreciate it if the authors could provide details of the backbone algorithms and details for training and execution. For example, is the algorithm Q-learning based or Actor-critic based; what are the information available to the actions and critics respectively?

2- the experimental results are not sufficient. I expect to see the comparisons to some multi-agent algorithms, such as PIC[ Liu2019]. Dependent on the answers to Question 1, it is possible that more baselines need to be added.


**Summary Of The Paper:**

This work considers the symmetries properties in cooperative multi-agent systems.
By decomposing global symmetries into local transformations, this work introduces a multi-agent equivariant policy network based on this factorization. Empirical results show that  on symmetric multi-agent problems, distributed execution of globally symmetric policies improves data efficiency compared to non-equivariant baselines.


**Summary Of The Review:**

This paper is in general well written and easy to follow. The method is well motivated. My major concern is that the paper lacks comparisons to existing algorithms that are either permutation invariant or not. And the information of training and execution is missing.

---

> ### Author Response · Authors · 2021-11-22
> **Reply to Reviewer xEvs**
>
> We thank the reviewer for their comments and suggestions. We are glad to hear that the reviewer considers the paper clearly written and easy to follow, and the method well motivated.
>
> **Training details**
>
> We thank the reviewer for pointing out the missing policy training. We train our networks with a standard PPO implementation, which will adjust the coefficients of the weight matrices in the network. The output of the network results in a policy and value predicted per agent. The information available to the actors and critics is their local information and the information received from neighbors. We have added the missing information to the paper.
>
> **Baseline clarification**
>
> The main goal of our experiments is to evaluate and understand the potential of equivariance in distributed symmetric multi-agent systems in isolation without real world interference. We do compare to other multi-agent methods, such as a permutation equivariant network (“standard MPN”), akin to PIC. We have clarified that our current baselines are multi-agent approaches in the paper. Thank you.

---

### Official Review · Reviewer_XJk1 · 2021-11-01

**Correctness:** 4
**Technical Novelty And Significance:** 3
**Empirical Novelty And Significance:** 4
**Recommendation:** 8
**Confidence:** 2

**Main Review:**

Strengths:

- The observations made about symmetry in MMDPs are reasonable. The intuition behind these observations are well illustrated by giving a concrete example of a cooperative traffic light control system.

- The network structure is well-explained and appears to be able to better leverage local symmetry.

- While the code is not yet made public, the experiments section in the Appendix is sufficiently detailed and the results appear to be replicable.

Weaknesses:

- I am unfamiliar with the area, but it seems the baselines compared against are limited. Since the paper is empirical in nature, could the authors comment more on what are other related works on applied MMDP that could be used for the same task? Can the authors show that the proposed method outperforms other algorithms, that perhaps only make use of graphs or permutation symmetries?

**Summary Of The Paper:**

The paper introduces the notion of symmetry in MMDPs and proposes an empirical algorithm that could leverage symmetry in MMDPs. By decomposing distributable global symmetries into local symmetries, the paper proposes a new neural network structure that empirically outperforms existing MPN models.

**Summary Of The Review:**

Overall, the paper introduces an interesting observation on certain real-world MMDP tasks and proposes a novel network structure that performs well under two different tasks, one with real-world data and the other with simulations.

---

> ### Author Response · Authors · 2021-11-22
> **Reply to Reviewer XJk1**
>
> We thank the reviewer for their positive review and suggestions regarding baselines. We are glad that the reviewer considers the paper and observations well-explained and well illustrated.
>
> We agree that comparing to a graph or permutation-only symmetry baseline is important. This corresponds to our standard MPN baseline, which is indeed permutation equivariant/graph based, but does not consider rotational symmetries.
> We have clarified the baseline in the experiments (labels “standard MPN”). Thank you.

---

### Official Review · Reviewer_Z7zL · 2021-11-02

**Correctness:** 3
**Technical Novelty And Significance:** 2
**Empirical Novelty And Significance:** 3
**Recommendation:** 6
**Confidence:** 3

**Main Review:**

The problem this paper considers is very interesting, but the proposed method confuses the reviewer.

(1) The novelty of this paper.  Is Multi-Agent MDP Homomorphic Network a simple combination of Graph Neural Networks and Symmetrizer? What are the main challenges to exploiting symmetry in MARL?

(2) The writing is not clear. Why is there a $|G|$ in $\phi_e$'s output in Sec. 4.3.2? Can the authors use framework diagrams instead of codes in Appendix D to show the architectural details?  These obscurities are very confusing, especially when some definitions in the code are not defined in this paper. The reviewer strongly recommends the authors not omit some details (although the details can be found in *MDP Homomorphic Networks: Group Symmetries in Reinforcement Learning*).

But in general, the idea of exploiting symmetry in MARL is timely and interesting. The reviewer hopes the authors could provide a running example to explain how the proposed method works, including training and executing. And the author also should clarify what the main contributions are.

**Summary Of The Paper:**

This paper tries to exploit the global symmetries in the joint state-action space of cooperative multi-agent systems. To allow distributed execution using only local information, this paper introduces Multi-Agent MDP Homomorphic Networks that use a factorization of global symmetries into symmetries on local observations and local interactions. They further compare the proposed networks with non-equivariant distributed networks, and show that global equivariance improves data efficiency on two games.

**Summary Of The Review:**

This paper is not well-written and very confusing. Refinement is needed.

---

> ### Author Response · Authors · 2021-11-22
> **Reply to Reviewer Z7zL**
>
> We thank the reviewer for their comments and suggestions for improving the paper. We are glad to hear that the reviewer considers the problem studied here timely and very interesting.
>
> **Novelties**
>
> Multi-Agent MDP Homomorphic Networks bring the following novelties: (i) a factorization that decomposes global symmetries into local transformations; (ii) a framework for distributed execution under global equivariance; and (iii) a novel policy network that exploits global symmetries while allowing for distributed execution. With our approach, we are now able to address one of the main challenges for exploiting symmetry in MARL, namely benefiting from global symmetries without requiring a centralized controller. This is not possible in a simple combination of graph networks with a symmetrizer. We have clarified our novelties in the introduction of the paper.
>
> **Writing clarity**
>
> We thank the reviewer for pointing out where we could have been clearer. $|G|$ represents the number of group elements, aka the size of the group. We have clarified in Section 4.3.2. For improving clarity regarding the architecture, we thank the reviewer for their guidance and we have updated the appendix accordingly, including a diagram of the general network architecture.
> Following the reviewer’s suggestions, we have also added an explanation of our training setup to the Experiments section, and have clarified the main contributions in the introduction.
>
>
> We thank the reviewer for helping to improve the clarity and writing of the paper. We have made updates throughout the paper.

---

### Official Review · Reviewer_Jvcy · 2021-11-02

**Correctness:** 4
**Technical Novelty And Significance:** 3
**Empirical Novelty And Significance:** 3
**Recommendation:** 6
**Confidence:** 3

**Main Review:**

### Strengths:
- The authors identify symmetries which arise in multi-agent settings and identify ways to exploit them.
- The type of symmetry identified in Figure 1 is more interesting than those in MDP Homomorphic networks.  It is similar to the type of actions considered in Steerable CNN, in which the group acts on both a spatial base space and on a fiber space as well.
- The method is well-motivated and well-designed in that the architecture reflects and enforces both the global permutation symmetries and local rotational symmetries the authors identify in the problem.
- The experiments show a clear improvement in data efficiency which is quite important in RL.
- The scope section in the conclusion does a nice job identify limitations in the current work and pointing the way to interesting future work.

### Weakness/Limitations:
- The symmetry in Figure 1, while interesting, is not really outside the framework of MDP with symmetry.  By simply assuming full communication and making a single centralized agent, we can still consider this symmetry by permutation/rotation as an MDP symmetry of a single-agent MDP with symmetries.
- The test domains are both fairly simple and feel a bit contrived.  In particular the traffic light scenario is only symmetric because the traffic lights have exactly symmetric lay out.  City grids notwithstanding, actual road segments are usually distinguishable in the real world.
- The authors point out they only use discrete groups, but the work is actually more limited than that.  They only use permutation actions and the discrete rotation group $C_4$.  Since the method is formulated to work with many groups, I’d like to see it tested with more groups.  In particular, for image data, $C_4$ is a best case for equivariant networks.  Using different groups would involve more complexity and potential downsides.
- It seems like data augmentation only involved randomly transforming each sample.  In practice, often additional samples are used with many transformations.  The argument could be made that this is not really less data efficient although it takes longer to train.  It might be good to compare to augmentation with additional samples.

### Questions
- The action in equation (6) is only each local state and edge attribute.  So the new local state at node $i$ is computed in terms of the old local state at node $i$.   But the permutation action as in Figure 1 also permutes the local states.  Is some permutation missing from this defn?
- If the MPN has M layers and M is sufficiently large, then how distributed or constrained is communication practically?  For example, in the traffic light setting, I assume the lights can talk to their neighbors, so after M layers of an MPN, can’t all lights effectively communicate?  So isn’t that the same as a centralized control?  From the definition of MMDP, it seems logical that the information sharing constraint is per action step not per layer.
- What is the advantage of using Symmetrizer?  It seems a basis of linear $G$-intertwiners here would not be difficult to derive analytically (as most authors do).
- In general, an MPN is fully permutation equivariant and, moreover, equivariant to permutation of node labels independent of actions or other features.  So using an MPN would seem to give too much symmetry.  In the traffic light setting, you want to be equivariant only to simultaneous cyclic permutations of agents and rotations of features.  However, I believe the edge features do the job of breaking the full symmetry of MPN and give exactly the symmetry you want.  Is this the case?  If so, can it be made more explicit or proved?
- I assume the non-linearities are pointwise and applied only to regular representations?  If so, it should be explained.
- Since only $C_4$ symmetry is used, I assume the Equivariance error is very small for the fully equivariant method.  Did you empirically verify this?  Also, how small is the equivariance error for the other baselines?

### Small Notes:

- Sec 3.1/ para 3: what is $\Delta(A)$?
- Sec 3.2/ para 3: $e$ is not defined.  Also it seems a bit strange to be explicit with $\cdot$ for composition but only use juxtaposition for the group action.
- Sec 3.2/ para 3: this is a nitpick, but it is not “the same group action” if it is on a different space
- Page 3 last para/ Page 4: The notion of “equivalent” state-action pairs doesn’t seem precisely defined to me.  In particular, it’s not clear to me what the sentence after Defn 2 is saying.
- Page 6, sentence after eqn 10: this is a nitpick, but linear combinations of equivariant functions are only equivariant if the output space is a group representation.  This would not hold for arbitrary equivariant functions between spaces with non-linear actions.
- Eqn 14.: Is $H_g$ defined?





**Summary Of The Paper:**

The authors give a generalization of MDP Homomorphic Networks (van der Pol, 2020) to multi-agent distributed settings.  This is not a special case of previous work as the agents are not assumed to be able to communicate freely.  The symmetry constraints of the single-agent case generalize to the multi-agent case with the twist that the group action also induces a permutation of the agents.  To create a neural network which matches this symmetry, the authors use a message-passing NN with equivariance constraints.  The method shows improved sample efficiency relative to networks which are only permutation equivariant trained with and without augmentation.  Only the discrete 90-degree rotation group is considered.



**Summary Of The Review:**

Overall, I tend towards accept.  The work appears to be sound and the method is well-motivated and well-designed.  I feel the extension from MDP Homomorphic Networks is not so large.  I believe the symmetries considered here may also be essentially considered as symmetries in the setting of van der Pol, 2020.  I feel the experiments are also somewhat limited.  They consider only the group $C_4$, compare only to versions of the proposed model, and the domains seem a bit contrived.  It is thus hard to evaluate how useful the method would be in practice.

---

> ### Author Response · Authors · 2021-11-22
> **Reply to Reviewer Jvcy**
>
> We thank the reviewer for their positive words on the setting, motivation and
> design of our approach, and for their suggestions to improve the paper.
>
> **Multi-agent symmetries and test domains**
>
> We agree with the reviewer that the theory of MDP with symmetries is the starting point of both MDP Homomorphic Networks and our paper. In the single agent framework, one needs centralized computation or at least high bandwidth communication. The key to our work is using such global symmetries in a setting that allows for distributed execution, i.e. without collecting all observations in a central location.
> Our test domains indeed abstract away a number of real world complexities. This is purposefully done, to allow us to evaluate - for the first time - the potential of equivariance in distributed symmetric multi-agent systems in isolation without real world interference.
>
> **On discrete groups**
>
> The goal of our work is to lay a foundation for incorporating group symmetries in distributed multi-agent systems and we can by design handle any discrete group. For our experiments, we have indeed focused on the most canonical discrete group in this subfield, namely C4. We believe the theory and initial empirical results provide valuable insights into symmetries for multi-agent systems and are excited for future large-scale investigations with increased real world and group complexity. We have added the justification for the experimental scope to the Experiments section. Thank you.
>
> **Additional augmentation baseline**
>
> Based on the reviewer’s suggestion, we have implemented a second data augmentation baseline with more augmentations, where every state is augmented with all its rotations in the group, and we transform both states and policies. Initial results on traffic light control show that this baseline outperforms the current augmentation baseline, while our equivariant approach performs better. We have updated Figure 6 and we are currently performing further tuning and evaluation on other benchmarks for in a final paper version. Thank you.
>
> **Remaining questions**
>
> _Equation 6_
>
> Thank you for pointing out the missing permutation in equation 6. We have added it to the paper.
>
> _Distributed control setting_
>
> In multi-agent systems, distribution is preferred over centralized control for two reasons: (i) having a centralized controller is simply not always feasible (for instance it could require expensive communication infrastructure to guarantee to transmit all sensor data in realtime), making centralized execution difficult. Examples of such settings include traffic light control, remote drone swarms, or other settings with fast but limited communication bandwidth and/or range. (ii) For many problems, centralized control induces an exponential growth of the action space. We therefore seek to find a distributed solution while still benefiting from global symmetries.
>
> _Advantages of the Symmetrizer_
>
> The main advantage of using the Symmetrizer is in terms of implementation ease. Our approach is agnostic to the method of finding the intertwiners.
>
> _Permutations and edge features_
>
> Indeed MPNs are symmetric to permutation as is normally the case in graph networks, where the identities of agents don’t matter as long as the state and edge features are the same. In addition, our setting requires global rotational equivariance which consists of combined rotations and permutations of the observations. This global rotational equivariance is enforced using equivariance constraints on the state and edge functions.
>
> _Non-linearities and equivariance error_
>
> We use pointwise non-linearities and regular representations. We have clarified this in the paper. We have confirmed that the equivariance error for the equivariant networks is within machine precision.
>
> _Small notes_
>
> Thank you for pointing out the small writing issues, we have addressed them.

---

### Decision · Program_Chairs · 2022-01-20

**Decision:**

Accept (Poster)

**Comment:**

The paper proposes some interesting ideas about decomposing the global symmetries of multi-agent MDP to local symmetries using a method called the Homomorphic Networks. The paper is well-writing and can be followed easily. However, there are a number of weaknesses of the paper. Below, we list some of the outstanding ones.

(1) Both some of the reviewers and the AC find there are some clarity issues in the paper, for instance, it is hard to see why using keeping relative communication and local transformation would guarantee symmetry (it would be good to show this with a theorem); it is hard to understand the algorithmic structure due to the use of codes instead of diagram in Appendix D, especially when some definitions in the code are not defined in this paper; the AC also finds it is hard to understand how communication is performed during both training and executing. After the rebuttal, the structure of the paper has been improved but we encourage the authors to keep improving the presentation.

(2) Another weakness is the experiment implementations, which the reviewers found were a bit too simple and contrived -- the traffic light example may not be symmetric in the real world. It might be good to demonstrate the effectiveness of the setting in a more realistic setting -- perhaps show that the method also works with a slight violation of symmetry.

That being said, the contribution of the paper remains significant, and the AC recommends borderline but slightly leaning toward acceptance.